# The mother's risk of premature death after child loss across two centuries

Unnur A Valdimarsdóttir[1,2,3†*], Donghao Lu[1,2,3,4†], Sigrún H Lund[5], Katja Fall[3,6], Fang Fang[3], Þórður Kristjánsson[5], Daníel Guðbjartsson[5,7], Agnar Helgason[5,8‡], Kári Stefánsson[5,9‡*]

[1]Center of Public Health Sciences, Faculty of Medicine, University of Iceland, Reykjavik, Iceland; [2]Department of Epidemiology, Harvard TH Chan School of Public Health, Boston, United States; [3]Department of Medical Epidemiology and Biostatistics, Karolinska Institutet, Solna, Sweden; [4]Channing Division of Network Medicine, Brigham and Women's Hospital, Harvard Medical School, Boston, United States; [5]deCODE Genetics, Reykjavik, Iceland; [6]Clinical Epidemiology and Biostatistics, School of Medical Sciences, Örebro University, Örebro, Sweden; [7]School of Engineering and Natural, Sciences, University of Iceland, Reykjavik, Iceland; [8]Department of Anthropology, University of Iceland, Reykjavik, Iceland; [9]Faculty of Medicine, School of Health Sciences, University of Iceland, Reykjavik, Iceland

**\*For correspondence:**
unnurav@hi.is (UAV);
kstefans@decode.is (KS)

[†]These authors contributed equally to this work
[‡]These authors also contributed equally to this work

**Competing interests:** The authors declare that no competing interests exist.

**Abstract** While the rare occurrence of child loss is accompanied by reduced life expectancy of parents in contemporary affluent populations, its impact in developing societies with high child mortality rates is unclear. We identified all parents in Iceland born 1800–1996 and compared the mortality rates of 47,711 parents who lost a child to those of their siblings (N = 126,342) who did not. The proportion of parents who experienced child loss decreased from 61.1% of those born 1800–1880 to 5.2% of those born after 1930. Child loss was consistently associated with increased rate of maternal, but not paternal, death before the age of 50 across all parent birth cohorts; the relative increase in maternal mortality rate ranged from 35% among mothers born 1800–1930 to 64% among mothers born after 1930. The loss of a child poses a threat to the survival of young mothers, even during periods of high infant mortality rates.
DOI: https://doi.org/10.7554/eLife.43476.001

## Introduction

The death of a child is arguably one of the most stressful events that a parent can experience. Grief is the natural emotional response to such loss and has been described across human populations (*Lindemann, 1944*; *Bowlby, 1980*; *Parkes, 2014*; *Boyden et al., 2014*; *Einarsdóttir, 2004*). Intense or prolonged grief can induce behaviours and physiological states that are detrimental to health and survival. Thus, child loss has been associated with elevated risks of psychiatric symptoms (*Kreicbergs et al., 2004*), psychiatric hospitalizations (*Li et al., 2005*) and sickness absence (*Wilcox et al., 2015*), cardiovascular disease (*Li et al., 2002a*; *Li et al., 2003a*), and some cancers (*Fang et al., 2011*; *Huang et al., 2013*; *Li et al., 2002b*). Studies also suggest that parents, particularly mothers, who lose a child face increased risk of death (*Li et al., 2002a*; *Li et al., 2003a*; *Levav et al., 1988*; *Li et al., 2003b*; *Qin and Mortensen, 2003*; *Agerbo, 2005*; *Rostila et al., 2012*; *Rostila et al., 2015*; *Espinosa and Evans, 2013*; *Harper et al., 2011*; *Chen et al., 2012*; *Schorr et al., 2016*), both from suicide and natural causes. The aforementioned studies were all conducted in affluent developed populations, where infant mortality rates are low and life expectancy is

high. In contemporary developing nations, infant mortality rates (number of deaths during first year per 1000 live births) >50 are common, whereas in developed nations they are usually lower than eight (*United Nations, 2015*). Parent-child attachment in pre-industrial and developing populations has been a matter of debate, with some historians arguing for parental reluctance to invest emotions in children where mortality rates are high, while others argue that parents' attachment and grief after child loss is similar across human populations, past and present (*Magnússon, 2010*; *Woods, 2003*). One way to shed light on this issue is to study the impact of child loss on parent mortality in the same population before and after a demographic transition that dramatically altered infant mortality rates.

Here, we examine the association between child loss and parent mortality in Iceland from 1817 until 2015, a period that spans a drastic transition from a poor agricultural society to an affluent industrial society. One marker of this drastic change is that infant mortality rates in Iceland (per 1000 births) dropped from one of the highest known values in history, 238 in 1881–1885 (*Jónsson and Magnússon, 1997*), to one of the lowest, 2.07 in 2005–2010 (*Statistics Iceland, 2019*). We used a comprehensive genealogical database to assess whether the relative rate of parent mortality after child loss changed during this transition, and to determine whether this rate was further affected by factors such as parent's sex and age at loss, the number of children as well as the age of the lost child. In line with some previous studies (*Li et al., 2003b*), we used parent mortality as a measure of the cumulative health response to the emotional and physiological strain of child loss. While mortality obviously does not capture the entire range of such reactions, it has the advantage of being reliably recorded at different times in a genealogical database.

## Results

We followed all Icelandic parents, born 1800 or later, from the birth of their first child (earliest from 1817) until death or to the end of 2015 (see further in Materials and methods). From 1817 to 2015, 64 044 parents lost at least one child (*Supplementary file 1* - Table 1). The proportion of parents who experienced child loss decreased from 61.1% among parents born 1800–1880 to 5.2% among parents born 1931–1996. To adjust for potential confounding by socioeconomic status or other familial factors that might affect the risk of mortality, our primary analyses were focused on the 47,711 parents who lost a child and had at least one sibling who did not lose a child (N = 126,342). Compared to their siblings who did not lose a child, parents who lost a child were more likely to be women, have more children, be younger at first childbirth and older at cohort entry, and have a longer follow-up time (p<0.05; *Table 1*). Similar patterns were observed for mothers and fathers who lost a child, respectively (*Supplementary file 1* - Table 2).

### Parent mortality rate after child loss across two centuries

We used a conditional Cox proportional hazards model to contrast the mortality rates of parents who lost at least one child to their respective siblings who did not (*Figure 1*). Throughout the entire study period, we observed an elevated hazard ratio (HR) of overall mortality rate among parents who lost a child (1.04; 95% confidence interval [CI] 1.02–1.06), which was mostly driven by maternal (HR 1.07; 95% CI 1.04–1.10) rather than paternal mortality rates (HR 1.02; 95% CI 0.99–1.05; p-for-difference=0.0096). *Figure 1* shows that there are no obvious temporal trends in the hazard ratios, except for an increase in mortality rate after child loss for parents born after 1930 (p-for-difference = 0.002 when compared to parents born 1800–1930). Similar results were obtained from the population-based matched cohort and a subset of this cohort with identifiable siblings, separately, (*Figure 1—figure supplement 1*), albeit with greater HRs revealing the importance of the sibling-controlled design to adjust for unmeasured familial confounding factors. The HRs changed minimally after exclusion of parent-child pairs with the same date of death (*Figure 1—figure supplement 2*).

### Child loss and premature maternal mortality rate

As the average life expectancy of Icelanders increased considerably from 1817 to the present, we performed separate analyses of parent mortality before and after the age of 50 years (*Figure 2*). Before the age of 50, mothers who lost a child experienced increased mortality (HR 1.36; 95% CI 1.27–1.45) during the whole study period with the HRs consistently elevated across all seven birth cohorts (*Figure 2a*). For mothers, the relative mortality rate increase ranged from 35% (born 1800–

Table 1. Descriptive characteristics of parents born from 1800 to 1996 who lost a child by death during their life course, and their siblings who did not lose a child, N (%).

| | Parents with loss | Parents without loss | P for difference |
|---|---|---|---|
| Total number | 47 711 | 126 342 | - |
| Sex | | | |
| Female | 25 125 (53) | 63 328 (50) | 3.65e-21 |
| Male | 22 586 (47) | 63 014 (50) | |
| Total number of children | 276 819 | 531 736 | - |
| Number of children, mean (SD) | 5.80 (3.19) | 4.21 (2.71) | 0 |
| Number of children | | | |
| 1 | 1775 (4) | 16 012 (13) | 0 |
| 2–4 | 17 766 (37) | 64 908 (51) | |
| 5–9 | 21 947 (46) | 39 019 (31) | |
| 10+ | 6223 (13) | 6403 (5) | |
| Age at first child birth, mean (SD) | 25.55 (5.09) | 27.16 (6.11) | 0 |
| Age at matching[*], mean (SD) | 38.76 (15.39) | 37.35 (14.60) | 8.84e-67 |
| Age at matching[*] | | | |
| 13–30 | 17 963 (38) | 52 434 (42) | 1.84e-70 |
| 31–50 | 19 980 (42) | 51 348 (41) | |
| 51–75 | 8123 (17) | 19 480 (15) | |
| 76+ | 1645 (3) | 3080 (2) | |
| Age of deceased child, mean (SD) | 9.44 (14.95) | - | - |
| Length of follow-up, mean (SD) | 31.79 (18.36) | 25.45 (18.03) | 0 |
| Age at death | 70.81 (16.66) | 71.80 (16.20) | 1.40e-21 |

* Age at loss of first child for parents who lost a child; the same age for their siblings who did not lose a child or the age when the siblings became parents, whichever came later.

DOI: https://doi.org/10.7554/eLife.43476.002

1930) to 64% (born after 1930), yet was not statistically significantly different between these two cohorts (p=0.0641). In contrast, no such excess mortality was observed for fathers who lost a child in any of the birth cohorts (HR for entire observation period 1.04; 95% CI 0.97–1.11). Interestingly, the excess maternal mortality rates before the age of 50 after child loss seem limited to loss of a child younger than 18 years, while the elevation in paternal mortality rates are more notable after loss of an adult child (*Table 2*). The difference between mothers and fathers in excess mortality rate before the age of 50 is particularly marked after loss of infants, representing 59% of child losses during the study period. These results were robust to the exclusion of mothers who died within a week after giving birth (*Figure 2—figure supplement 1*), making it unlikely that the higher mortality rate in mothers is explained by delivery or postpartum complications that resulted in the deaths of both newborn and mother (*Nour, 2008*).

## Later-life mortality rate after child loss

After the age of 50 and across the entire study period, neither mothers (HR 1.02; 95% CI 0.99–1.05) nor fathers (HR 1.01; 95% CI 0.98–1.05) suffered increased mortality rate after child loss. Separately analyzed by time since loss and demographic characteristics, we did not observe significantly increased mortality in response to child loss for parents after 50 years of age (*Supplementary file 1*

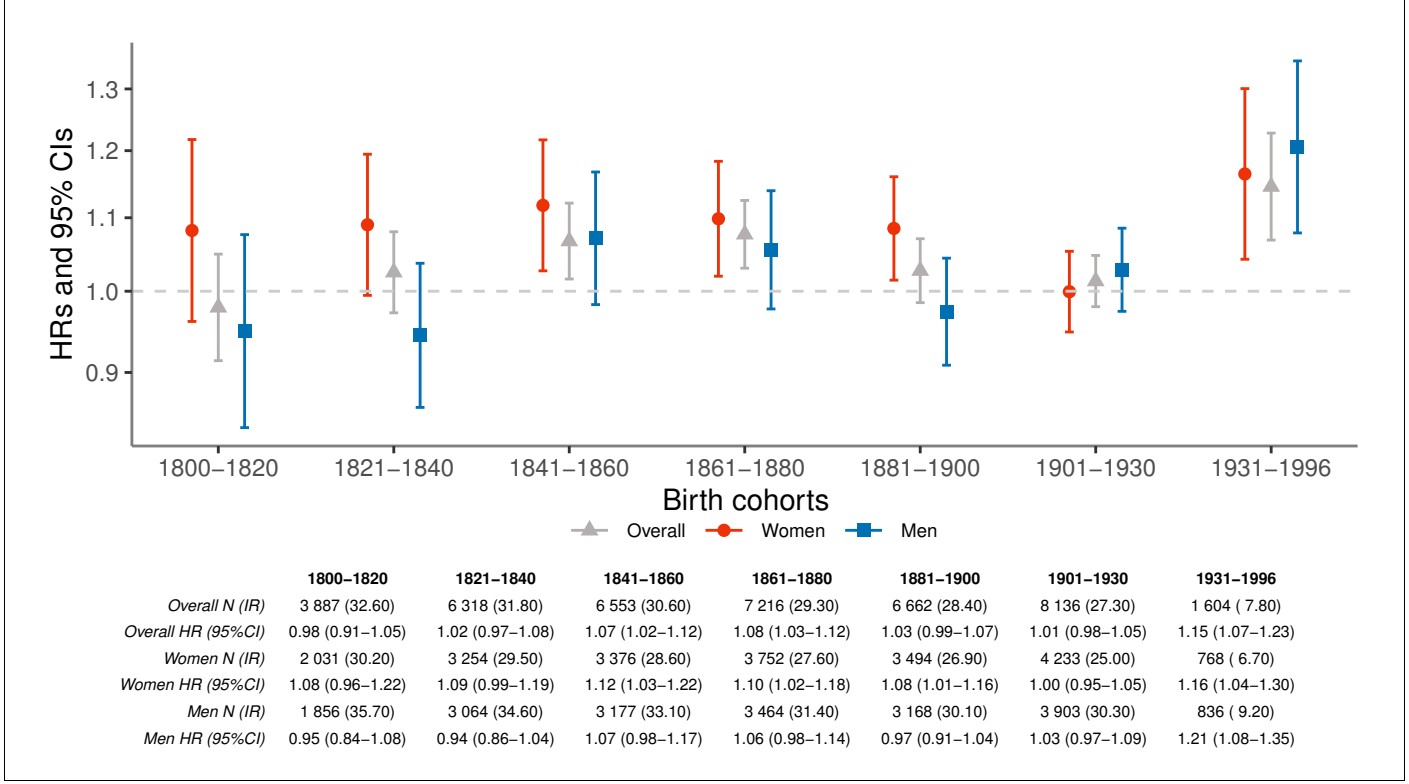

**Figure 1.** Hazard ratios (HRs) and 95% confidence intervals (CIs) of parental mortality after loss of a child by birth cohorts (every 20 years until 1900, 1901–1930, and 1931–1996), in the sibling cohort. We estimated HRs from stratified Cox proportional hazards model using age as underlying time scale. We stratified by sibling groups and additionally adjusted for birth year and sex. IR, incidence rate, per 1000 person-years.

DOI: https://doi.org/10.7554/eLife.43476.004

The following figure supplements are available for figure 1:

**Figure supplement 1.** Hazard ratios (HRs) and 95% confidence intervals (CIs) of parental mortality after loss of a child among all parents and parents with identifiable siblings (those subsequently included in the sibling cohort) in the population-based matched cohort.

DOI: https://doi.org/10.7554/eLife.43476.005

**Figure supplement 2.** Hazard ratios (HRs) and 95% confidence intervals (CIs) of parental mortality after loss of a child by birth cohorts (every 20 years until 1900, 1901–1930, and 1931–1996) in the sibling cohort, excluding parents who died on the same day as the child.

DOI: https://doi.org/10.7554/eLife.43476.006

- Table 3). However, when broken down into the seven smaller birth-cohort periods, we observed significantly increased mortality in parents older than 50 who lost a child and were born after 1930 (HR 1.11; 95% CI 1.03–1.20; p=0.0063), particularly among fathers (HR 1.21; 95% CI 1.08–1.37; p=0.0009; *Figure 2b*). Curiously, this recent increase in mortality after the age of 50 years is associated with child loss that mostly occurred before the age of 50 (*Figure 2—figure supplement 2*). As a result, we postulate that the recent increase in parent mortality rate after the age of 50, particularly in fathers, may be due to an increased risk of long-term social isolation after child loss. In support of this notion, we indeed found very high mortality rates for parents born after 1930 who lost a child and had no living children at the age of 50, regardless of the sex of deceased child (*Supplementary file 1* - Table 4).

In a sensitivity analysis, restricting to parent-sibling pairs born within five years from the parents who lost a child, comparing same-sex siblings, or adjusting for the number of children alive at time of matching, we observed similar estimates of parental mortality rates after child loss (*Figure 2—figure supplements 3–5*). The only notable difference was that, in the birth cohort 1931–1996, the impact of child loss on maternal mortality after 50 years of age became statistically significant. We further restricted unexposed siblings to those who were already parents at the age when the index sibling lost a child, which also yielded similar results (*Figure 2—figure supplement 6*).

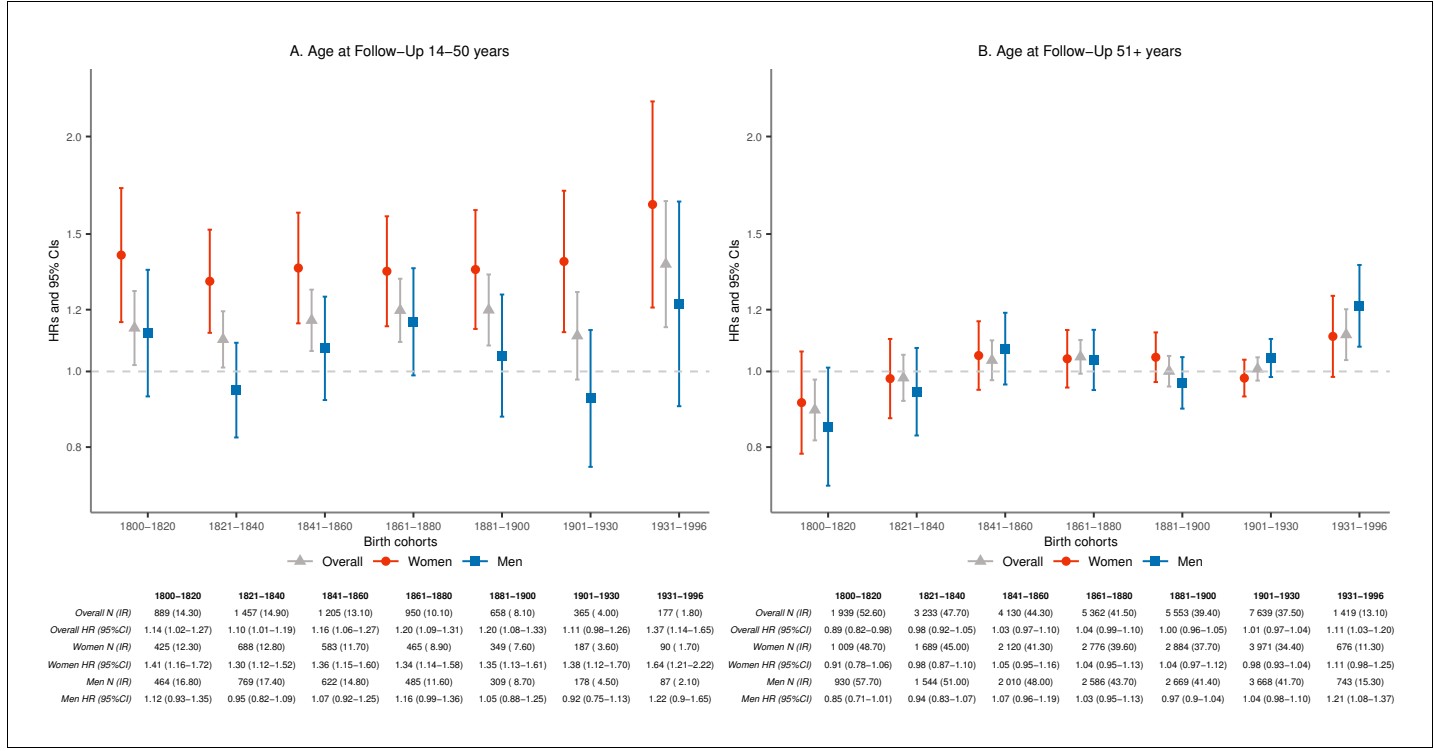

**Figure 2.** Hazard ratios (HRs) and 95% confidence intervals (CIs) of parental mortality after loss of a child, by age bands at follow-up (age 14–50 and 51 +) and birth cohorts (every 20 years until 1900, 1901–1930, and 1931–1996), in the sibling cohort. We stratified by sibling groups and additionally adjusted for birth year and sex in Cox proportional hazards model. IR, incidence rate, per 1000 person-years.

DOI: https://doi.org/10.7554/eLife.43476.007

The following figure supplements are available for figure 2:

**Figure supplement 1.** Hazard ratios (HRs) and 95% confidence intervals (CIs) of maternal mortality after loss of a child, by age bands at follow-up, excluding mothers dying within 1 week after giving birth, in the sibling cohort.

DOI: https://doi.org/10.7554/eLife.43476.008

**Figure supplement 2.** Hazard ratios (HRs) and 95% confidence intervals (CIs) of parental mortality after loss of a child among parents born during 1931–1996 and followed from age 51 onward, by age at loss, in the sibling cohort.

DOI: https://doi.org/10.7554/eLife.43476.009

**Figure supplement 3.** Hazard ratios (HRs) and 95% confidence intervals (CIs) of parental mortality after loss of a child, by age bands at follow-up (age 14–50 and 51+) and birth cohorts (every 20 years until 1900, 1901–1930, and 1931–1996), among the siblings born 5 years before or after the index parents.

DOI: https://doi.org/10.7554/eLife.43476.010

**Figure supplement 4.** Hazard ratios (HRs) and 95% confidence intervals (CIs) of parental mortality after loss of a child, by age bands at follow-up (age 14–50 and 51+) and birth cohorts (every 20 years until 1900, 1901–1930, and 1931–1996), by restricting to same-sex siblings.

DOI: https://doi.org/10.7554/eLife.43476.011

**Figure supplement 5.** Hazard ratios (HRs) and 95% confidence intervals (CIs) of parental mortality after loss of a child, by age bands at follow-up (age 14–50 and 51+) and birth cohorts (every 20 years until 1900, 1901–1930, and 1931–1996), in the sibling cohort with additional adjustment for number of alive children at matching.

DOI: https://doi.org/10.7554/eLife.43476.012

**Figure supplement 6.** Hazard ratios (HRs) and 95% confidence intervals (CIs) of parental mortality after loss of a child, by age bands at follow-up (age 14–50 and 51+) and birth cohorts (every 20 years until 1900, 1901–1930, and 1931–1996), by restricting to unexposed siblings who were already parents at the same age as when the index parents lost a child.

DOI: https://doi.org/10.7554/eLife.43476.013

## Discussion

The study period spans two centuries of drastic demographic transition in Iceland, from a poor developing nation with a particularly high infant mortality rate, to a modern affluent nation with one of the lowest rates of infant mortality in history. Although we observed a more modest elevation in

**Table 2.** Hazard ratios (HRs) and 95% confidence intervals (CIs) of premature mortality after loss of a child among young parents followed up to age 50, by time since loss and demographic characteristics, compared to their siblings.
We stratified by sibling groups and additionally adjusted for birth year and sex. IR, incidence rate, per 1000 person-years.

| | Overall | | Women | | Men | |
|---|---|---|---|---|---|---|
| | N (Crude IR) | HR (95% CI) | N (Crude IR) | HR (95% CI) | N (Crude IR) | HR (95% CI) |
| Time since loss | | | | | | |
| 0–4 years | 1711 (10.2) | 1.19 (1.11–1.27) | 907 (10.3) | 1.52 (1.35–1.71) | 804 (10.0) | 0.96 (0.85–1.07) |
| 0–4 years* | 1595 (9.5) | 1.10 (1.03–1.18) | 832 (9.4) | 1.40 (1.24–1.58) | 763 (9.5) | 0.89 (0.80–1.01) |
| 5–9 years | 1145 (7.9) | 1.13 (1.05–1.23) | 508 (6.5) | 1.22 (1.06–1.41) | 637 (9.5) | 1.02 (0.90–1.16) |
| 10–19 years | 1654 (8.2) | 1.15 (1.07–1.23) | 764 (6.7) | 1.33 (1.18–1.50) | 890 (10.0) | 1.09 (0.97–1.22) |
| 20–39 years | 388 (6.2) | 1.12 (0.98–1.29) | 214 (5.3) | 1.28 (1.03–1.59) | 174 (7.9) | 1.16 (0.91–1.48) |
| Child's age at loss | | | | | | |
| 0 | 3357 (8.9) | 1.14 (1.08–1.20) | 1672 (8.0) | 1.43 (1.31–1.57) | 1685 (9.9) | 0.94 (0.86–1.03) |
| 1–5 | 2016 (9.4) | 1.16 (1.09–1.24) | 964 (8.1) | 1.26 (1.13–1.41) | 1052 (11.0) | 1.12 (1.00–1.25) |
| 6–17 | 269 (6.7) | 1.17 (0.99–1.38) | 130 (5.6) | 1.31 (0.99–1.74) | 139 (8.3) | 1.32 (1.00–1.73) |
| 18+ | 66 (5.4) | 1.44 (1.05–1.97) | 22 (2.8) | 0.96 (0.54–1.71) | 44 (10.3) | 1.99 (1.24–3.20) |
| Number of alive children at loss | | | | | | |
| 0 | 2390 (9.7) | 1.20 (1.13–1.28) | 1135 (8.3) | 1.39 (1.24–1.56) | 1255 (11.4) | 1.15 (1.04–1.28) |
| 1–3 | 2793 (8.4) | 1.12 (1.06–1.18) | 1379 (7.5) | 1.37 (1.25–1.51) | 1414 (9.5) | 0.98 (0.89–1.07) |
| 4+ | 525 (8.1) | 1.12 (0.99–1.26) | 274 (7.3) | 1.18 (0.97–1.44) | 251 (9.1) | 0.94 (0.77–1.14) |
| Sex of the lost child | | | | | | |
| Female | 3075 (8.6) | 1.14 (1.08–1.20) | 1483 (7.5) | 1.29 (1.17–1.41) | 1592 (10.0) | 1.06 (0.97–1.16) |
| Male | 2633 (9.2) | 1.17 (1.10–1.24) | 1305 (8.2) | 1.44 (1.30–1.59) | 1328 (10.4) | 1.01 (0.92–1.11) |
| Age at loss | | | | | | |
| 13–30 | 3459 (8.5) | 1.13 (1.08–1.19) | 1815 (7.5) | 1.33 (1.22–1.45) | 1644 (10.1) | 1.02 (0.93–1.12) |
| 31–40 | 1945 (9.5) | 1.19 (1.12–1.27) | 873 (8.7) | 1.43 (1.27–1.61) | 1072 (10.2) | 1.04 (0.93–1.15) |
| 41–50 | 304 (8.7) | 1.14 (0.98–1.32) | 100 (6.1) | 1.11 (0.81–1.52) | 204 (11.0) | 1.12 (0.90–1.40) |
| Age at first childbirth | | | | | | |
| 13–21 | 988 (6.0) | 1.20 (1.09–1.32) | 653 (5.4) | 1.29 (1.13–1.47) | 335 (7.7) | 1.15 (0.95–1.40) |
| 22–24 | 1507 (8.5) | 1.09 (1.01–1.17) | 829 (8.0) | 1.27 (1.12–1.44) | 678 (9.2) | 0.97 (0.84–1.11) |
| 25–27 | 1520 (10.3) | 1.20 (1.11–1.30) | 665 (9.5) | 1.35 (1.17–1.56) | 855 (11.1) | 1.08 (0.96–1.23) |
| 28+ | 1693 (10.8) | 1.16 (1.08–1.24) | 641 (9.9) | 1.55 (1.34–1.80) | 1052 (11.3) | 1.01 (0.91–1.13) |

* A sensitivity analysis excluding parents dying on the same day as their child.
DOI: https://doi.org/10.7554/eLife.43476.003

overall parent mortality rates after child loss before 1930 compared to thereafter, the most striking finding is the consistent increase in premature maternal deaths in response to child loss throughout the 199 years of observation.

This study leverages a unique nationwide data source of genealogies with virtually complete follow-up over a 199 year period spanning an extreme change in indicators of societal development, for example infant and child mortality rate. Although based on an unusually large set of data, our study has several limitations that should be noted. First, we have no information about the causes of death for children or parents. Such data could be used to help verify the causal relationship between deaths of children and parents and to uncover patterns in the causes of death among parents (e.g. suicide or cardiovascular deaths) after child loss. Some impact of reverse causation, for example the ill-health of parents increasing the mortality risk of their children, cannot be ruled out. However, the health and welfare of children was even more dependent on their parents, particularly fathers, in pre-demographic transition (pre-welfare state times) societies. The relatively stable effect estimates of parent mortality rates after child loss across calendar periods and the consistently lower impact of

child loss on paternal mortality argues against a strong influence of reverse causality. Second, we did not have direct information about the socioeconomic status of parents. However, as socioeconomic status tends to cluster in families, our sibling-controlled analysis is likely to adjust for much of the confounding due to this source. Third, it is possible that the fragmented registration of births in earlier times (particularly in the case of infant deaths) resulted in misclassification of some parents who had truly lost a child and, thus, somewhat conservative estimates of excess parent mortality rates in earlier times. Last, in the sibling cohort, parents who lost a child differed somewhat from those that did not lose a child with respect to sex, number of children, age at first birth, age at entry to the cohort, length of follow-up time and age at death. We therefore made every effort to adjust for these covariates in our analyses. We adjusted for age at follow-up and sex in all analyses and presented separate results for the different birth cohorts. We further performed subgroup analyses by age of parents at loss (or matching), parent age at first childbirth, follow-up time and number of children alive at loss, and demonstrated that the association between child loss and parent mortality is observed across the range of these covariates. Indeed, parents who lost a child may have to live longer to experience such an event, however, such survivor bias (if any) due to a selection of survivors to the exposed group would rather yield conservative estimates of parent mortality after a child loss. On the other hand, the siblings used for comparison that were not yet parents at the age as when the index parents lost a child may have a survival advantage, which could yield exaggerated estimates of parent mortality after child loss. However, we observed similar results in a sensitivity analysis that was restricted to unexposed siblings who were already parents at the same age as when the index parents lost a child.

The large increase in the rate of premature maternal mortality after child loss in 19[th] century Iceland suggests that mothers formed strong emotional bonds with their children – in spite of very harsh conditions and a strong expectation of child loss. This is consistent with accounts of intense grieving among mothers who lost a child in a contemporary developing society (*Einarsdóttir, 2004*). In contrast, the paternal mortality response to child loss is much weaker throughout the observation period and only statistically significant for older fathers (i.e. parent mortality after 50 years of age) born after 1930.

What could account for this stable sex difference in parent mortality rate after child loss? One possibility is a difference in the attachment of mothers and fathers to their children, which could result in a greater emotional response of mothers and concomitant psychological and physiological consequences. Throughout the 199-year study period, mothers may have had greater opportunity than fathers to form early emotional attachment to their infants through gestation and breastfeeding. Although historical accounts suggest that breastfeeding may have been rarer in 19th to early 20th century Iceland (*Garðarsdóttir, 2005*) than in the present, it does not seem to have prevented mothers from forming strong emotional bonds with their children as illustrated by the elevated risks of maternal death after child loss. It is possible that females are born with a tendency to form stronger emotional bonds with children than males. Although our results cannot disentangle the impact of nature and nurture in this context, it is interesting to note that the mortality response of fathers to child loss before the age of 50 has not significantly increased despite the greater contribution of fathers to the postnatal care of children in recent decades (*Huerta et al., 2014*).

Another possible explanation for our findings is a sex difference in the emotional response to traumatic events in general. Women have been reported to have a greater risk of mental disorders after exposure to emotional trauma than men (*Tolin and Foa, 2006*). The intensity of such responses may result in a multitude of other mental and physical ailments which in turn can affect the risk of mortality through, for example suicide (*Wilcox et al., 2009*; *Panagioti et al., 2012*; *Gradus et al., 2010*) and cardiovascular disease (*Song et al., 2019*; *Edmondson et al., 2013*; *Gradus et al., 2015*). Previous studies in contemporary developed populations have reported evidence for greater risk of psychiatric hospitalizations (*Li et al., 2005*) and mortality (*Li et al., 2003b*) after child loss in mothers than in fathers. Interestingly, an opposite sex difference is seen for mortality rate after spousal loss, where mortality rate elevation is more pronounced among widowers than widows (*Moon et al., 2011*; *Shor et al., 2012*). However, this difference may in part be due to a greater dependence of older males on social and emotional support provided by spouses. A similar mechanism is indicated in our results, namely the increased mortality of older parents after child loss in more recent times.

Our results show that even in the poor agricultural population of 19th century Iceland, with one of the highest infant mortality rates measured in humans, there was a considerable impact of child loss on premature maternal mortality rates. This contradicts claims that mothers in poor cultures adapt to high infant mortality rates with reduced emotional investment (as reviewed in *Woods, 2003*). Indeed, one implication of our findings is that child loss is likely to constitute a major threat to the survival of mothers in societies with high infant mortality rates. Taken together these findings highlight the importance of the mother-child bond and the extensive health threats that arise when it is broken.

## Materials and methods

### Data sources and study participants

The data source for the current study is the deCODE Genetics genealogy database, containing information on kinship, dates of birth and death of Icelandic population largely since settlement (*Gudbjartsson et al., 2015*). The database was constructed by compiling information from church records, various annals and censuses from 1703 to 1930 as well as the contemporary registers of the total population. The database contains an almost complete record of the ancestors of contemporary Icelanders back to around 1650 and, after 1880, is based on death records from a geographically comprehensive set of parish records. In terms of linkages between parents and offspring, the completeness of the genealogical database across the entire observation period is 99%. The database has been extensively evaluated for the accuracy of such linkage using genetic data (*Sun et al., 2012*; *Helgason et al., 2015*; *Halldorsson et al., 2019*).

We conducted a historical cohort study of parents who were born from January 1st 1800 through December 31st 1996 and were living in Iceland. In total 323,510 individuals were eligible for inclusion; 27,704 individuals who emigrated after 1906 were not included in the study population. Individuals entered the cohort from the birth year of their first child (earliest from 1817), and were followed until their own death or through 2015, whatever happened first. We excluded 12,233 (3.8%) parents who were lost to follow-up.

This study was reviewed by the National Bioethics Committee and Data Protection Authority of Iceland (Approval No. VSN-12–125 and VSN-16–156) that in accordance to Icelandic law (Act no. 44/2014 on scientific research within the health sector and Act no. 77/2000 on the protection of privacy in processing of personal data) permit large-scale, population-based register studies, such as this one, to be carried out without informed consent.

### Ascertainment of exposure to child loss and covariates

All parents entering the cohort were followed for exposure to child loss, indicated by the date of a child's death being earlier or on the same day as the parent's date of death. Information about mortality in the Icelandic genealogy database are obtained from parish records, from 1735 to 1970, and thereafter from the Population Register held by Statistics Iceland. We further obtained information about the parents' sex and age when having their first child, the total number of children and their sex. In cases where parents experienced child loss, we obtained information about parents' age when losing their first child and the child's age at the time of death.

### Follow-up and ascertainment of parental mortality

We followed all parents from the year of entry to the study (when first becoming parents). The person-time of parents was defined as unexposed from entry to the cohort until the loss of their first child, death, or end of the observation period (end of 2015), whichever happened first. Person-time was defined as exposed (for parents who lost a child) from the date of their first child loss until death or end of the observation period (end of 2015).

### Population-based matched cohort

Since life expectancy has changed dramatically across the study period, we performed population-based matched cohort analysis, based on 64,050 parents who lost a child during follow-up, who were matched on birth year (±1 year) and sex to parents who did not lose a child. To this end, we randomly selected three parents (born before 1920) or five parents (born 1920 or later) who had not

lost a child at the time when the index parent lost a child (i.e. the reference time). Six parents who lost a child but could not be matched to any parents without loss were excluded. The resulting data set comprised 64,044 parents who lost a child and 218,824 matched parents who did not lose a child. We followed all participants from the reference time until death or December 31st, 2015, whichever came first.

### Siblings-based cohort

To allay the concerns of potential confounders shared by family members (such as socioeconomic status, lifestyle and genetic components), we restricted our primary analysis to a sibling-based cohort. Of note, although the siblings used as reference in this analysis did not lose a child, they did lose a niece or nephew which, if truly influencing the rate of mortality, may lead to conservative estimates. Among 64,044 parents with loss, 54,532 (85.1%) had at least one sibling defined as sharing the same biological mother. Each parent with loss was paired with three available siblings on average (range 2–16) who had not lost a child at the same age when the index parent lost a child (i.e. the reference age), whereas 6821 parents were excluded because they only had siblings who had experienced child loss. We followed all participants from the reference age, or the age when the siblings became parents, whichever came later, until the age at loss of a child, death, or December 31st, 2015, whichever occurred first. Specifically, siblings who did not lose a child started to contribute to the follow-up at the same age as the parent who lost a child or at a later age when they became parents. The sibling-based cohort comprised 47,711 (74.5%) parents who lost a child and 126,342 matched siblings without loss.

### Statistical analysis

We first compared the characteristics of parents who lost a child (exposed) with those who did not (unexposed) across birth cohorts for the following covariates: sex, mean age at loss, mean child's age at the time of death and total number of children. In this analysis, we used a *chi-square* test for the categorical and a *t-test* for the continuous covariates. We then calculated the crude mortality rates (deaths per 1000 person-years) in both exposed and unexposed cohorts, overall and stratified by parental sex, birth cohorts, and/or age at follow-up.

Although we performed parallel analyses on the population-based matched cohort, the primary analysis was focused on the siblings-based cohort which provides a better control of unmeasured confounders shared by siblings. We used conditional Cox proportional hazard regression models to estimate hazard ratios (HRs) and 95% confidence intervals of mortality for exposed parents relative to unexposed parents, while stratifying on the matched pairs in analyses of population-based matched cohort or sibling groups in analyses of siblings-based cohort. We used time since the reference time and age at follow-up as the underlying timescale for population-based matched cohort and sibling-based cohort, respectively. All models were additionally adjusted for birth year for the population-based matched analysis (because ±1 birth year was allowed in matched pairs), and adjusted for birth year and parental sex in the sibling-based analysis. Of note, we did not match on parental sex in the sibling-based cohort but adjusted for it in all models and thereby accounting for the effect of sex on mortality rate across the sibling groups. Matched pairs or sibling groups who were invariant with regard to the outcome were included, although they did not contribute to the estimates. Guided by the timing of the dramatic societal change and to ensure sufficient statistical power for each period, the same analyses were performed for seven birth cohorts separately (every 20 years from 1800 to 1900, 1901–1930, and 1931–1996). Because of the better control of potential confounders, we only performed the subsequent analyses in the sibling-based cohort.

To investigate potential bias due to common external causes of death (e.g. pandemics or accidents taking the lives of both parents and children) we performed a sensitivity analysis excluding all parent-child pairs with identical date of death. Due to differential impact across birth cohorts, the subsequent analyses were always separated by birth cohorts. All analyses were performed on men and women separately.

As the average life expectancy of Icelanders increased considerably from 1817 to the present, we performed separate analyses of parent mortality before and after the age of 50 years, that is 13-50 and ≥51 years. To reduce the potential differences between siblings we, in sensitivity analyses, restricted to parent-sibling pairs born within 5 years from the parents who lost a child (N = 96,041,

55.2%) as well as to unexposed siblings who were already parents at the same age as when the index parents lost a child (N = 135,719, 78.0%). To further address the potential influences of parent sex and number of children, in another sensitivity analysis, we only compared same-sex siblings and additionally adjusted for number of alive children at the time of loss/matching, respectively.

Parent mortality before and after age of 50 years was further analyzed in subgroups of age of the deceased child (0, 1–5, 6–17, or ≥18 years old), number of children alive at loss (0, 1–3, or ≥4), sex of the deceased child, parental age at loss (13–30, 31–40, or 41–50 years old), and parental age at first childbirth (13–21, 22–24, 25–27, or ≥28 years).

The excess risk of mortality beyond age 50 among parents born 1931–1996 was further explored with respect to the age of parents when the loss occurred (13–30, 31–50 or ≥51 years); as well as sex of deceased child and number of children by age 50.

All statistical analyses were performed using R, version 3.3.1. Survival analysis was performed with the packages 'survival' and graphs were plotted with the package 'ggplot2'. All R-codes used in the primary analyses are available (Source Code File 1).

## Acknowledgements

The study is funded by the Icelandic Research Fund- RANNIS (Grant of Excellence; nr: 163362–051) and the European Research Council (StressGene; nr: 726413). We are grateful that the librarians from the Karolinska Institutet University Library provided professional help on literature search.

## Additional information

### Funding

| Funder | Grant reference number | Author |
|---|---|---|
| RANNIS | 163362-051 | Unnur Valdimarsdóttir |
| European Research Council | 726413 | Unnur Valdimarsdóttir |

The funders had no role in study design, data collection and interpretation, or the decision to submit the work for publication.

### Author contributions

Unnur A Valdimarsdóttir, Conceptualization, Supervision, Funding acquisition, Methodology, Writing—original draft, Writing—review and editing; Donghao Lu, Formal analysis, Methodology, Writing—original draft, Writing—review and editing; Sigrún H Lund, Formal analysis, Writing—review and editing; Katja Fall, Fang Fang, Writing—review and editing; Þórður Kristjánsson, Project administration, Writing—review and editing; Daníel Guðbjartsson, Supervision, Methodology, Writing—review and editing; Agnar Helgason, Conceptualization, Data curation, Supervision, Writing—original draft, Writing—review and editing; Kári Stefánsson, Conceptualization, Data curation, Writing—review and editing

### Author ORCIDs

Unnur A Valdimarsdóttir (iD) https://orcid.org/0000-0001-5382-946X
Donghao Lu (iD) https://orcid.org/0000-0002-4186-8661

### Ethics

Human subjects: This study was reviewed by the National Ethics Committee and Data Protection Authority of Iceland (Approval No. VSN-12–125 and VSN-16–156).

### Decision letter and Author response

Decision letter https://doi.org/10.7554/eLife.43476.018
Author response https://doi.org/10.7554/eLife.43476.019

## Additional files

### Supplementary files

• Source code 1. R script used for the primary analyses.
DOI: https://doi.org/10.7554/eLife.43476.014

• Supplementary file 1. Descriptive characteristics of parents born from 1800 to 1996 who lost a child by birth cohorts in the population-based matched cohort, N (%).
DOI: https://doi.org/10.7554/eLife.43476.015

• Transparent reporting form DOI: https://doi.org/10.7554/eLife.43476.016

### Data availability

The data used in this study are compiled in the Genealogy database (Íslendingabók) at deCODE Genetics and are available for review on site, on request. The use of these data is in accordance with Icelandic law and with permission of the Icelandic Bioethics Committee. Therefore, the authors cannot make the dataset publicly available. On the other hand, interested researchers can obtain data access upon approvals by the Icelandic Bioethics Committee (https://www.vsn.is/en) and by contacting the corresponding authors.

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
