## [Decision Letter]

**Acceptance summary:**

This is an interesting and valuable analysis that uses the historic registry data in Iceland to ask whether losing a child increases the risk of death for the parents. As the authors state, several papers have shown loss of a child is associated – in observational studies – with a higher risk of parental death, particularly for mothers. However, the novelty in this study is using a sibling comparison design over a 200-year time frame with a large sample size. Hence they are able to answer the question "is the phenomenon of interest – increased risk of parental death following the death of a child – particular to industrialized human populations in which child death is quite rare." The answer from this paper is clearly 'no' – even in a context in which child death affects more than half of parents, the authors detect a raised risk of death among mothers and (in a more specific context) fathers.

**Decision letter after peer review:**

Thank you for submitting your article "The mother's risk of premature death after child loss across two centuries" for consideration by *eLife*. Your article has been reviewed by two peer reviewers, and the evaluation has been overseen by M. Dawn Teare as the Reviewing Editor and Eduardo Franco as the Senior Editor. The following individuals involved in review of your submission have agreed to reveal their identity: Susan C Alberts (Reviewer #1).

The reviewers have discussed the reviews with one another and the Reviewing Editor has drafted this decision to help you prepare a revised submission.

Essential revisions:

1) There are some puzzling features of this study that it would be helpful to understand better. Parents who lost a child and had more children, were younger at first birth, were more likely to be women and had longer follow-up that those who did not lose a child (Table 1). Was the difference in% women for parents with and without loss significant? If so, please add to the text. More generally, please explain these differences between the parents with and without loss, and consider whether the differences are an artefact of the design/implementation that might bias the estimates.

2) The difference in age between parents with loss and their siblings could be because follow-up time started when the child died for the parents with loss, and started for the sibling when the sibling became a parent. As such, for the sibling the time between their nephew/niece dying and the birth of their first child might be "immortal" time (1) that gives the sibling a survival advantage. For example, if the parent and their childless sibling both died a week after the child died, then the sibling would not be eligible for the study, making the surviving eligible siblings more strongly selected survivors than the parents with loss. Please consider whether such an issue could affect the estimates and amend the method or add a limitation explaining this point.

3) Please explain the adjustment for sex in the sibling design in more detail. Was the sibling design adjusted for sex composition of the sibling pairs, i.e., both female, both male, etc., because men and women have children at different ages, giving a different exposure time till age 50 years?

4) Given the above concerns about the handling of exposure time, would it be helpful to add a twin comparison possibly with exposure time handled in a different way?

5) Number of children could be a confounder (common cause of exposure and outcome) or a mediator, because a higher number of children affects parental survival. Obviously, the more children in a family the greater risk of a child dying. Alternatively, parents might have a 'replacement' child after a death. Would it be possible to distinguish between these possibilities and conduct the analyses accordingly? Would a marginal structural model help? Alternatively, it would be helpful to see estimates adjusted for number of children in the family.

6) The authors allude to an important point in the Discussion when they say "…we have no information about the causes of death for children or parents. Such data could be used to help verify the causal relationship between deaths of children and parents and to uncover patterns in the causes of death among parents (e.g. suicide or cardiovascular deaths) after child loss." This seems a highly salient point: Indeed, a causal relationship in the other direction is quite possible, and mutual causal arrows are possible. That is, child death may be greatly increased in cases where parents are unwell or in poor condition – implying a causal arrow from parent health to child death. Conversely, the risk of parental death after child death may be particularly high for parents in poor health. The authors need to lay out these alternative causal pathways more explicitly. Is there any data that could provide insight into these questions?

7) It would be good to know a bit more about the completeness (or lack thereof) of the dataset and whether the authors used any specific techniques to deal with this incompleteness. For instance, The authors make the statement "it is possible that the fragmented registration of births in earlier times (particularly in the case of infant deaths) resulted in misclassification of some parents who had truly lost a child and, thus, somewhat conservative estimates of excess parent mortality rates in earlier times." And "The database contains an almost complete record of the ancestors of contemporary Icelanders back to around 1650 and, after 1880, is based on death records from a geographically comprehensive set of parish records." It is difficult to get a full picture of dataset completeness from this relatively scanty information. What proportion of records are incomplete? A table with information about various types of missing data would be advisable.

8) It would be helpful if the discussion focused more on the strengths and weaknesses of the study.

[Editors' note: further revisions were requested prior to acceptance, as described below.]

Thank you for resubmitting your work entitled "The mother's risk of premature death after child loss across two centuries" for further consideration at *eLife*. Your revised article has been favorably evaluated by a Senior Editor and a Reviewing Editor, and two reviewers.

The manuscript has been improved but there is one remaining issue that needs to be addressed before acceptance. Reviewer #2 still has concerns about survivor bias. Comparing people who survived at least one child with those who did not may result in selection bias. Adjusting for potential confounding variables will not protect against this. Essentially the people who have child loss may have to live longer, which may obscure the effect of losing a child. The authors need to elaborate on this issue. This could be done by explaining why this is not a problem, performing an additional comparison or highlighting the issues under limitations. We transcribe below the reviewers' comments.

Reviewer #1:

The authors have thoroughly and clearly addressed the concerns and completed the essential revisions. I believe the paper is ready for publication and will make a strong contribution to the literature.

Reviewer #2:

This is an interesting study looking at the relation of offspring death with survival over 200 years in a unique data set.

My previous concern with this study was that the people who had lost a child are different from the comparison group who did not lose a child, which might cause bias, due to some form of survivor bias, i.e., selection bias. Specifically, the parents with loss are more likely to be women, younger at first birth, to have more children and to have longer follow-up. The authors have responded by saying that they addressed these issues by adjustment or stratification, which are means of dealing with confounding not selection bias.

Essentially, the study appears to be comparing people who survived at least one of their children with people who did not survive any of their children. It is easier to survive at least one of your children if you are a woman (because women live longer than men), are younger when the children are born, if you have more children and are followed up for longer which explains why the people with child loss are more likely to be women, younger at first birth, with more children and to have longer follow-up.

I remain concerned as to whether comparing people who survived at least one child with those who did not gives the effect of child loss on survival. Essentially the people who have child loss may have to live longer, which may obscure the effect of losing a child. Of course, it is possible that these issues do not matter, but it would be helpful if the authors could explain why not, or amend accordingly.

---

## [Author Response]

Essential revisions:1) There are some puzzling features of this study that it would be helpful to understand better. Parents who lost a child had more children, were younger at first birth, were more likely to be women and had longer follow-up that those who did not lose a child (Table 1). Was the difference in% women for parents with and without loss significant? If so, please add to the text. More generally, please explain these differences between the parents with and without loss, and consider whether the differences are an artefact of the design/implementation that might bias the estimates.

Given the large sample size, any differences between parents who did vs. did not lose a child is likely to be statistically significant even when the absolute difference is small. We have now added P for difference for all comparisons in Table 1 and revised the text accordingly. The P for difference across parental groups was <0.05 for all covariates, including: parents’ sex, number of children, age at first birth, age at entry to the cohort, length of follow-up time and age at death.

Although we note a statistically significant difference between sibling groups on all covariates we have directly addressed each of these factors in our analyses and, therefore, are confident that they do not account for our results. We adjusted for age at follow-up (as the underlying timescale) and sex in all analyses and present results for each of the studied birth cohorts throughout. In Table 2, we further present the association between child loss and premature parent mortality rate stratified by parental age at loss/matching and follow-up time, and sub-grouped by the number of children alive at loss. These results are not consistent with the notion that our findings are driven by differences between the groups on the abovementioned factors. Although varying somewhat, the association between loss and survival is evident across the range of these covariates. The considerable differences in average follow-up time between parents who did vs. did not lose a child (31.79 vs. 25.45 years, respectively) is likely explained by the parents who did not lose a child at the time of matching but later encountered a child loss and were then censored as end of follow-up in that matching set. This is relatively common in the earlier birth cohorts due to the higher mortality rates for children in that period. This concern is partly alleviated by Figure 2, where the follow-up of parents is restricted to 50 years across all birth cohorts. Moreover, the results in Table 2 do not suggest a considerable variation in parent mortality across extended periods of follow-up time from child loss/matching. This further argues against the possibility that the difference of average follow-up length between two groups may account for our findings.

Modified manuscript text:

Results section: “Compared to their siblings who did not lose a child, parents who lost a child were more likely to be women, have more children, be younger at first birth and older at cohort entry, and have a longer follow-up time (P<0.05; Table 1).”

Discussion section: “Last, in the sibling cohort, parents who lost a child differed somewhat from those that did not lose a child with respect to sex, number of children, age at first birth, age at entry to the cohort, length of follow-up time and age at death. We therefore made every effort to adjust for these covariates in our analyses. We adjusted for age at follow-up and sex in all analyses and presented separate results for the different birth cohorts. We further stratified analyses by age of parents at loss (or matching), follow-up time and number of children alive at loss, and demonstrated that the association between child loss and parent mortality is observed across the range of these covariates.”

Materials and methods: “In this analysis, we used a chi-square test for the categorical and a t-test for the continuous covariates.”

See also Table 1.

2) The difference in age between parents with loss and their siblings could be because follow-up time started when the child died for the parents with loss, and started for the sibling when the sibling became a parent. As such, for the sibling the time between their nephew/niece dying and the birth of their first child might be "immortal" time (1) that gives the sibling a survival advantage. For example, if the parent and their childless sibling both died a week after the child died, then the sibling would not be eligible for the study, making the surviving eligible siblings more strongly selected survivors than the parents with loss. Please consider whether such an issue could affect the estimates and amend the method or add a limitation explaining this point.

We agree with the reviewer that the sibling design needs further clarification in the manuscript. First, in order to be at risk of losing a child, one needs to have one. Thus, our analysis of index persons who lost a child and their siblings only considers follow-up time where both of these groups were parents. We started to follow the siblings who already had a child from the same age as their index-sibling when he/she lost a child. If the siblings did not have any children at the age when the index-persons lost a child, these were not included in the analysis until they subsequently had one, and then only from the age when they became parents. Indeed, we agree with the reviewer that the time between the sibling’s nephew/niece dying and the birth of their first child is “immortal” and was therefore not included in our analysis and should not bias our estimates. Indeed, both groups (individuals with or without child loss) need to survive long enough to become parents to be considered for analysis. The reviewer notes that the parents who lost a child are a bit further into his/her life history than their siblings who are still childless at the time of the loss. However, the population cohort – where even higher hazard ratios are observed – does not have that problem. Please also refer to the results of our sensitivity analysis below (Point #4), restricting the sibling cohort to unexposed siblings born within 5 years of the parents who lost a child.

We have now clarified this important issue in the Materials and methods section.

Modified manuscript text:

“Specifically, siblings who did not lose a child started to contribute to the follow-up at the same age as the parent who lost a child or at a later age when they became parents.”

3) Please explain the adjustment for sex in the sibling design in more detail. Was the sibling design adjusted for sex composition of the sibling pairs, i.e., both female, both male, etc., because men and women have children at different ages, giving a different exposure time till age 50 years?

We did not restrict the sibling sets to all females or all males since that would inevitably have further raised the bar for selection of siblings. Instead, we matched a mother who lost a child to all her eligible siblings with children, including both her sisters and brothers, who had not lost a child at the same age. We adjusted for sex in the sibling-based analysis which accounts for the effect of sex on mortality rate across sibling sets. Moreover, we used attained age as the underlying time scale and therefore the age difference was also controlled for. We have now further clarified the sibling-based analysis in the Materials and methods.

Modified manuscript text:

“Of note, we did not match on parental sex but adjusted for it in all models and thereby accounting for the effect of sex on mortality rate across the sibling groups.”

4) Given the above concerns about the handling of exposure time, would it be helpful to add a twin comparison possibly with exposure time handled in a different way?

Unfortunately, a twin-based comparison would be underpowered due to the low number of pairs. However, we have now conducted a sensitivity analysis restricting to 96,041 (55.2%) siblings born five years before or after the index parents. The results remain largely unchanged, as presented in Figure 2—figure supplement 5.

Modified manuscript text:

Results section: “In a sensitivity analysis, restricting parent-sibling pairs born within five years from the parents who lost a child or adjusting for the number of children alive at time of matching, we observed similar estimates of parental mortality rates after child loss (Figure 2—figure supplements 5-6). The only notable difference was that, in the birth cohort 1931-1996, the impact of child loss on maternal mortality after 50 years of age became statistically significant.”

Materials and methods: “In a sensitivity analyses, we restricted to parent-sibling pairs born within five years from the parents who lost a child (N=96,041, 55.2%) to reduce the potential differences between siblings.”

5) Number of children could be a confounder (common cause of exposure and outcome) or a mediator, because a higher number of children affects parental survival. Obviously, the more children in a family the greater risk of a child dying. Alternatively, parents might have a 'replacement' child after a death. Would it be possible to distinguish between these possibilities and conduct the analyses accordingly? Would a marginal structural model help? Alternatively, it would be helpful to see estimates adjusted for number of children in the family.

We agree with the reviewer that the number of children could be a confounder or a mediator in the association between child loss and parent mortality. This is why we did not adjust for it in our primary analysis. Instead, we performed a subgroup analysis by the number of children alive at the time of loss among parents who lost a child (Table 2), where we did not observe varying effect sizes across subgroups. In accordance with the reviewer’s note, we now present results from a sensitivity analysis, adjusting for the number of children alive at loss/matching. The results are consistent with those of the main analyses, except that maternal mortality after 50 years of age, in the birth cohort 1931-1996, becomes statistically significant (Figure 2—figure supplement 6). Adjusting for or matching on children born after loss (“replacement children”) is challenging, since the parents need to survive long enough to have them. This would introduce a survival bias. We have therefore opted not to adjust for the number of any post-loss/matching children in our analysis.

Modified manuscript text:

Results section:

In a sensitivity analysis, restricting to parent-sibling pairs born within five years from the parents who lost a child or adjusting for the number of children alive at time of matching, we observed similar estimates of parental mortality rates after child loss (Figure 2—figure supplements 5-6). The only notable difference was that, in the birth cohort 1931-1996, the impact of child loss on maternal mortality after 50 years of age became statistically significant.

Materials and methods: To further address the potential effect of number of children, in another sensitivity analysis, we additionally adjusted for number of alive children at the time of loss/matching.

6) The authors allude to an important point in the Discussion when they say "…we have no information about the causes of death for children or parents. Such data could be used to help verify the causal relationship between deaths of children and parents and to uncover patterns in the causes of death among parents (e.g. suicide or cardiovascular deaths) after child loss." This seems a highly salient point: Indeed, a causal relationship in the other direction is quite possible, and mutual causal arrows are possible. That is, child death may be greatly increased in cases where parents are unwell or in poor condition – implying a causal arrow from parent health to child death. Conversely, the risk of parental death after child death may be particularly high for parents in poor health. The authors need to lay out these alternative causal pathways more explicitly. Is there any data that could provide insight into these questions?

This is a valid point that requires further clarification. We have no data on causes of death while previous studies (e.g. Li et al., 2003) suggest that child loss in modern times (after 1980) increases the risk of maternal death, particularly of unnatural causes but also natural causes. We agree with the reviewer that some risk of reverse causation (parent’s ill health increasing risk of child’s death and subsequent death of parent) cannot be ruled out. However, if deteriorating parental health was strongly associated with the risk of child death, we might expect the association between child loss and parent mortality to be particularly evident during the first years of follow-up. However, the relatively stable association between child loss and parent mortality across follow-up times (as shown in Table 2) does not indicate a strong influence of reverse causality in these data. Indeed, families (mothers and children) were extremely financially dependent on fathers, such that if fathers died or were unable to work, then families were often separated and children fostered – which usually meant very hard work and diminished living conditions for these children. Thus, if parents' ill-health was a driving factor, we would expect a stronger association between child loss and father's mortality. However, it is indeed maternal mortality rate, and not paternal mortality rate after child loss, which is consistently elevated across all follow-up periods.

Modified manuscript text:

Discussion: “Some impact of reverse causation, e.g. the ill-health of parents increasing the mortality risk of their children, cannot be ruled out. However, the health and welfare of children was even more dependent on their parents, particularly fathers, in pre-demographic transition (pre-welfare state times) societies. The relatively stable effect estimates of parent mortality rates after child loss across calendar periods and the consistently lower impact of child loss on paternal mortality argues against a strong influence of reverse causality.”

7) It would be good to know a bit more about the completeness (or lack thereof) of the dataset and whether the authors used any specific techniques to deal with this incompleteness. For instance, The authors make the statement "it is possible that the fragmented registration of births in earlier times (particularly in the case of infant deaths) resulted in misclassification of some parents who had truly lost a child and, thus, somewhat conservative estimates of excess parent mortality rates in earlier times." And "The database contains an almost complete record of the ancestors of contemporary Icelanders back to around 1650 and, after 1880, is based on death records from a geographically comprehensive set of parish records." It is difficult to get a full picture of dataset completeness from this relatively scanty information. What proportion of records are incomplete? A table with information about various types of missing data would be advisable.

In terms of linkages between parents and offspring, the completeness of the genealogical database across the entire observation period is 99%. The proportion of individuals without identifiable one or both parents in the database is: 2.4% in individuals born 1800-1880, 0.6% in individuals born 1881-1930, and 1.5% in individuals 1931-1996. Moreover, the database has been extensively evaluated for the accuracy of such linkages using genetic data – (Helgason et al., 2015, Sun et al., 2012, Halldorsson et al., 2019)

The individuals missing from the genealogical database are primarily children born prior to 1900, who died during the first few weeks of life and were not always recorded in censuses or parish records. This would lead to a misclassification of some parents, which in our study are considered as unexposed (parents that have not lost a child), but in fact should have been considered as exposed (parents who lost a child). We note that the impact of such misclassification on our results is in the form of a conservative bias. Thus, if there is a difference in survival between exposed and unexposed parents, such misclassification would only lead us to underestimate this difference and is therefore not a major concern for this study. We now provide this information in the Materials and methods section.

Modified manuscript text:

“In terms of linkages between parents and offspring, the completeness of the genealogical database across the entire observation period is 99%. The database has been extensively evaluated for the accuracy of such linkage using genetic data (40-42).”

8) It would be helpful if the discussion focused more on the strengths and weaknesses of the study.

Based on the points raised above we have now extended the discussion of limitations and moved that part to the beginning of the Discussion section.

[Editors' note: further revisions were requested prior to acceptance, as described below.]

The manuscript has been improved but there is one remaining issue that needs to be addressed before acceptance. Reviewer #2 still has concerns about survivor bias. Comparing people who survived at least one child with those who did not may result in selection bias. Adjusting for potential confounding variables will not protect against this. Essentially the people who have child loss may have to live longer, which may obscure the effect of losing a child. The authors need to elaborate on this issue. This could be done by explaining why this is not a problem, performing an additional comparison or highlighting the issues under limitations. We transcribe below the reviewers' comments.

Thank you for the opportunity for a second resubmission of our paper. Please find below our responses to the issues raised along with the corresponding modifications to the manuscript. We believe that we have now addressed reviewer #2’s main issue, i.e. survivor bias, and we hope you now find our manuscript acceptable for publication in *eLife*. Please do not hesitate to contact us with any remaining issues.

Reviewer #1:The authors have thoroughly and clearly addressed the concerns and completed the essential revisions. I believe the paper is ready for publication and will make a strong contribution to the literature.

Thank you for the positive comment on our work and our paper.

Reviewer #2:This is an interesting study looking at the relation of offspring death with survival over a 200 years in a unique data set.My previous concern with this study was that the people who had lost a child are different from the comparison group who did not lose a child, which might cause bias, due to some form of survivor bias, i.e., selection bias. Specifically, the parents with loss are more likely to be women, younger at first birth, to have more children and to have longer follow-up. The authors have responded by saying that they addressed these issues by adjustment or stratification, which are means of dealing with confounding not selection bias.Essentially, the study appears to be comparing people who survived at least one of their children with people who did not survive any of their children. It is easier to survive at least one of your children if you are a woman (because women live longer than men), are younger when the children are born, if you have more children and are followed up for longer which explains why the people with child loss are more likely to be women, younger at first birth, with more children and to have longer follow-up.I remain concerned as to whether comparing people who survived at least one child with those who did not gives the effect of child loss on survival. Essentially the people who have child loss may have to live longer, which may obscure the effect of losing a child. Of course, it is possible that these issues do not matter, but it would be helpful if the authors could explain why not, or amend accordingly.

The reviewer raises several important issues.

The reviewer remains concerned for a potential bias based on that “people who have child loss may have to live longer” than the comparison cohort of parents who do not experience a child loss. However, such type of bias, if any, caused by a selection of “survivors” to the exposed population would rather yield conservative estimates of the increase in parent mortality after a child loss. Also, as explained in more details below, we carefully designed our study to compare the two groups of parents, with and without a child loss, from the same age (i.e. the age when the loss of the child in one group occurred) whenever possible. Thus, both groups were at the same age at the start of follow-up (i.e., at the time of matching).

Thus, we remain convinced that our study design entails few, if any, possible conditions giving rise to survivor bias as described in the paper cited by the reviewer in the last round of review (2). We followed parents who lose a child starts immediately after child loss and, due to the fact that we use parental age as the underlying time scale, the follow-up of the matched parents not yet exposed to child loss starts at the same age/time point as for the exposed parents yielding identical conditions for ascertaining mortalities among these two compared groups. The sibling controls are followed from the same age as the parent that lost a child, or from the age when becoming a parent if they are not yet parents when their sibling lost a child; the latter, according to the definition above, may give rise to survivor bias. For this reason, we reran our sibling-based comparison, limiting it to unexposed siblings who were already parents at the age as the index sibling who lost a child. The results remained largely similar (if anything, the point estimates indicated even stronger associations) and we have added this sensitivity analysis to the revised manuscript (Figure 2—figure supplement 8).

Thus, we believe that the reviewer’s concern of survivor bias should be eliminated through our study design only considering person time of unexposed siblings who have reached the exact age at the index parent’s loss. In other words, both exposed and unexposed parent’s need to survive through the same age in order to be compared. Therefore, the amount of “immortal time” is similar across these groups. We have now clarified this issue by adding the following statements to the manuscript:

Modified manuscript text:

“We further restricted unexposed siblings to those who were already parents at the age when the index sibling lost a child, which also yielded largely similar results (Figure 2—figure supplement 8).”

“Indeed, parents who lost a child may have to live longer to experience such an event, however, such survivor bias (if any) due to a selection of survivors to the exposed group would rather yield conservative estimates of parent mortality after a child loss. […] However, we observed similar results when we in a sensitivity analysis restricted to unexposed siblings who were already parents at the same age as when the index parents lost a child.”

“To reduce the potential differences between siblings we, in sensitivity analyses restricted to parent-sibling pairs born within five years from the parents who lost a child (N=96,041, 55.2%) as well as to unexposed siblings who were already parents at the same age as when the index parents lost a child (N=135,719, 78.0%).”

In contrast to the abovementioned concern for selection (survivor) bias induced by the study design where certain individuals (or person time) are selected to or excluded from a study (3), we believe that the remaining points raised by the reviewer represent conditions of confounding but not selection bias (4). Our study is indeed a nationwide complete follow-up of all parents during a 199-year period; no individual is excluded from the study on the basis of sex, age at first childbirth or total number of children. As the reviewer correctly points out, these factors may be associated with parental survival and some of them are unequally distributed across the compared parental groups and therefore need to be accounted for in the analysis. We have indeed made extensive efforts to account for these factors in our analysis.

Sex is accounted for in the population analysis by matching on sex, i.e. the rates of mortality are only compared between pairs of exposed and unexposed mothers and then fathers, separately. To preserve statistical power, we did not match on sex in the sibling analysis while controlling for sex in all models. We have now performed additional analysis confined to sister- and then brother-pairs (discordant on child loss) yielding similar estimates in the primary analysis albeit with lower precision. These results have been added to the manuscript (Figure 2—figure supplement 6).

Modified manuscript text:

“In a sensitivity analysis, restricting to parent-sibling pairs born within five years from the parents who lost a child, comparing same-sex siblings, or adjusting for the number of children alive at time of matching, we observed similar estimates of parental mortality rates after child loss (Figure 2—figure supplements 5-7).”

“To further address the potential influences of parent sex and number of children, in another sensitivity analyses, we only compared same-sex siblings and additionally adjusted for number of alive children at the time of loss/matching, respectively.”

Other factors mentioned by the reviewer have appropriately, as potential confounders, been addressed in Table 2 in subgroup analyses. There we e.g. demonstrate that maternal mortality rates below the age of 50 are considerably elevated across parent groups with varying number of children, with the greatest excess observed after loss of an only child. In the last round of revisions, we further performed additional analyses controlling for the number of children at the time of matching revealing virtually identical, if anything stronger, results (Figure 2—figure supplement 7).

In accordance with the reviewer’s comment, we have now added a subgroup analysis of age at first child birth in Table 2, revealing that the association between child loss and maternal mortality rates below the age of 50 is present in every subgroup of parental age at first child birth.

Modified manuscript text:

“We further performed subgroup analyses by the age of parents at loss (or matching), parent age at first childbirth, follow-up time and number of children alive at loss, and demonstrated that the association between child loss and parent mortality was observed across the range of these covariates.”

“Parent mortality before and after age of 50 years was further analyzed in subgroups of age of the deceased child (0, 1-5, 6-17, or ≥18 years old), number of children alive at loss (0, 1-3, or ≥4), sex of the deceased child, parental age at loss (13-30, 31-40, or 41-50 years old), and parental age at first childbirth (13-21, 22-24, 25-27, or ≥28 years old).”

Finally, as we explained in the last round of revisions, the discrepancy between the parental groups in follow-up time is driven by the high incidence of child loss, particularly during 20th century. Our study design mimics a prospective study, where the investigators have at every time point of the observation no information on which parent will live or die, or lose a child. Therefore, parents were randomly selected as matched “control parents” while they had not lost a child and then censored from that control group at the time of child loss (if it occurred). Due to the high incidence of child loss, many parents were later censored from the control group at the time of child loss and assigned to a new set (now as index-individuals) yielding the relatively shorter follow-up time of parents not exposed to child loss relative to the exposed. Thus, the Cox model does not generate estimates of parental mortality rates based on comparison between two distinct groups of parents, exposed and unexposed to child loss, but on the basis of hazards during exposed and unexposed person time.

1) Levesque LE, Hanley JA, Kezouh A, et al. Problem of immortal time bias in cohort studies: example using statins for preventing progression of diabetes. BMJ 2010;340:b5087. doi: 10.1136/bmj.b5087 published Online First: 2010/03/17]

2) Lévesque LE et al. Problem of immortal time bias in cohort studies: example using statins for preventing progression of diabetes. BMJ, 2010;340:b5087.

3) Rothman KJ. Epidemiology: an introduction. Oxford university press; 2012: p126.

4) Newton J. Confounding is mistakenly called selection bias. BMJ, 1998. Accessed on https://www.bmj.com/rapid-response/2011/10/27/confounding-mistakenly-called-selection-bias